# Factors Contributing to Risk of Persistence of Positive and Negative Symptoms in Schizophrenia during Hospitalization

**DOI:** 10.3390/ijerph20054592

**Published:** 2023-03-05

**Authors:** Grzegorz Witkowski, Piotr Januszko, Michał Skalski, Anna Mach, Zbigniew Maciej Wawrzyniak, Ewa Poleszak, Bogdan Ciszek, Maria Radziwoń-Zaleska

**Affiliations:** 1Department of Descriptive and Clinical Anatomy, Medical University of Warsaw, 02-004 Warsaw, Poland; 2Department of Psychiatry, Medical University of Warsaw, 00-665 Warsaw, Poland; 3Faculty of Electronics and Information Technology, Warsaw University of Technology, 00-665 Warsaw, Poland; 4Laboratory of Preclinical Testing, Chair and Department of Applied and Social Pharmacy, Medical University of Lublin, 20-093 Lublin, Poland

**Keywords:** schizophrenia, cavum septi pellucidi, negative symptoms of schizophrenia, positive symptoms of schizophrenia, risk factors

## Abstract

The aim of the study was to evaluate factors that may contribute to the persistence of positive, negative and other psychopathological symptoms of schizophrenia. All patients were treated in general psychiatric wards between January 2006 and December 2017. The initial study sample comprised of the medical reports of 600 patients. The main, specified inclusion criterion for the study was schizophrenia as a discharge diagnosis. Medical reports of 262 patients were excluded from the study due to no neuroimaging scans being available. The symptoms were categorised into three groups: positive, negative, and other psychopathological symptoms. The statistical analysis comprised modalities such as demographic data, clinical symptoms, as well as neuroimaging scans linking them to a potential impact of sustaining the mentioned groups of symptoms during the period of hospitalization. The analysis revealed that statistically significant risk factors of persistence of the three groups of symptoms are the elderly age, the increasing toll of hospitalizations, suicidal attempts in medical history, a family history of alcohol abuse, the presence of positive, negative and other psychopathological symptoms on admission to the hospital, as well as the absence of cavum septi pellucidi (CSP). The study showed that addiction to psychotropic drugs and a family history of schizophrenia were more frequent in patients with persistent CSP.

## 1. Introduction

Schizophrenia is a mental condition, or rather a group of conditions [1], with widely varied symptoms and course patterns [2]. Psychiatry textbooks and other literature on the subject emphasize the fact that schizophrenia cannot be categorically classified as a chronic illness; however, the various authors seem to agree that in a considerable proportion of patients the illness is both chronic and progressive [3,4,5,6]. Thought disorders, abnormal perception, and changes in mood and affect co-exist in schizophrenic patients [7], along with physical anergia, avolition, and psychomotor retardation [8]. Cognitive functions, such as memory, concentration, and attention may remain intact during the initial years after the onset of schizophrenia; hence, they tend to be omitted in diagnostic guidelines [9]. Nevertheless, literature sources report them in a considerable proportion of patients, particularly many years after onset [9,10,11]. Individual patients with schizophrenia manifest symptoms from the three groups mentioned above in diverse ways and in various configurations.

Diagnostic classification systems, such as ICD-10, DSM-IV-TR, and DSM-5, which are used in clinical practice, incorporate symptoms otherwise known as positive or negative. The diagnostic criterion adopted in both classification systems is the presence of two (or more) of the following symptoms (delusions, hallucinations, disorganized speech, grossly disorganized or catatonic behavior, and negative symptoms, such as blunted affect or avolition), each of which is present most of the time. The various individual symptoms of schizophrenia are included in the diagnostic criteria of the classifications listed above and are characterized as first-rank (or more typical and diagnostically relevant) and second-rank (or less diagnostically relevant) symptoms [12].

The nosological status of schizophrenia would seem to well-established and stable, since the condition is included in two well-known diagnostic systems, e.g., the International Classification of Diseases, Tenth Revision (ICD-10) and the American Psychiatric Association’s Diagnostic and Statistical Manual of Mental Disorders, Fourth Edition, Text Revision (DSM-IV-Tr). However, treating schizophrenia as a distinct nosological entity is largely a matter of clinical convention; in fact, the condition is currently viewed from three different approaches. The monogenous approach is based on the idea that one disease is due to one cause. The polygenous approach is based on the idea that one disease is due to multiple causes. Finally, the heterogenous approach is based on the idea that diverse pathogenetic mechanisms may lead to several distinct disorders. The debate surrounding the nosological status of schizophrenia is a result of the condition’s abundant and diverse symptoms, heterogeneous course, and—most importantly—not entirely understood etiology and the parallel existence of several pathogenetic hypotheses. For many years, there have been discussions around whether schizophrenia should be treated as a separate and autonomous nosological entity or if it should rather be viewed as a ‘schizophrenia group’ or a ‘schizophrenia spectrum’.

The two well-known diagnostic systems use overall similar diagnostic criteria; the exclusion principles are also largely consistent. The key diagnostic criterion that is different between the two systems is the duration of schizophrenia symptoms. Establishing an ICD-10-based diagnosis requires the symptoms to have been present for at least one month, whereas a DSM-IV-TR-based diagnosis requires an at least a six-month history of symptoms (with the exception of the clinical situations mentioned above).

Clerambault [13] was the first to categorize the manifestations of schizophrenia into positive and negative symptoms, with the former later specified as including delusions, hallucinations, and disorganized behaviors, and the latter later specified as including emotional blunting, abulia, anhedonia, autism, and alogia.

Apart from the positive and negative symptoms of schizophrenia, the wide variety of psychopathological manifestations also includes other symptoms, which often accompany other mental conditions. These other, associated, but atypical symptoms of schizophrenia include mood disorders, anxiety, incoherent trains of thought, cognitive dysfunction, somatic disorders, and sleep disturbances.

For research purposes, these symptoms are usually presented in scales, such as the Positive and Negative Syndrome Scale for Schizophrenia (PANSS), the Scale for the Assessment of Positive Symptoms (SAPS), the Scale for the Assessment of Negative Symptoms (SANS) and the Clinical Assessment of Schizophrenic Syndromes (CASS) as three-factor models (apart from the negative symptoms, corresponding to ‘subtracted’ experiences, and the positive symptoms, corresponding to ‘added’ experiences, there is the other, or ‘mixed’, subgroup of symptoms) [12,14].

Despite many years of research into the pathogenesis of schizophrenia, it still remains unclear [15]. The etiological theories proposed over the years tend to complement one another rather than definitively explain the genesis of the illness [16]. Currently, the etiology of schizophrenia is generally accepted to be multifactorial, with the neurodevelopmental model as one of its key concepts [17,18,19,20,21]. This model is based on the concept that individuals with schizophrenia have a genetic predisposition to this illness and the development of their brain is disturbed by various factors during the prenatal, perinatal, and postnatal period, extending until adolescence [19].

Functional magnetic resonance imaging showed abnormalities in brain activity, primarily in the prefrontal cortex. The phenomenon of diminished activation of these areas of the brain during cognitive tests is known as hypofrontality.

As mentioned above, the neurodevelopmental model of schizophrenia combines several research hypotheses. The presence of structural abnormalities within the central nervous system may be due to certain anatomical variations, whose gross appearance in and of itself, without a broader clinical context, is not associated with any overt pathology. One such central-nervous-system structure that has certain documented anatomical variations is described below.

The septum pellucidum, which is a median-sagittally, or midsagittally, oriented structure within the forebrain, extends between the fornix and the genu and body of the corpus callosum separating the anterior horns of the lateral ventricles. The septum pellucidum consists of two thin, parallel laminae, separated by a slit-like space, known as the cavity of the septum pellucidum or the cavum septi pellucidi (CSP); (Figure 1 and Figure 2). 

Developmental abnormalities of midline prosencephalic structures have been linked with schizophrenia for a number of years. However, although the reduction in the long diameter of the interthalamic adhesion (adhesio interthalamica) and disturbances in the surface morphology of the orbitofrontal cortex seem well established in schizophrenic patients, the role of the CSP is still unclear. There are reports that negate the significance of this structure in schizophrenia, whereas some strongly support this idea [22,23].

These discrepancies focused our attention on the CSP cavum septi pellucidi in schizophrenic patients.

The laminae of the true septum pellucidum are made up of white matter, with the adjacent gray matter forming septal nuclei. The septal nuclei are associated with other parts of the limbic system (e.g., the hippocampus, the limbic cortex, and the amygdala). The function of the septum pellucidum itself, as an isolated structure, is unknown. Nonetheless, its normal development is known to be one of the factors determining the development of the forebrain [24]. An absent septum pellucidum is found in approximately 3 out of 100,000 of individuals [25]. The development of the septum pellucidum is associated with the development of the corpus callosum, and its absence results in the agenesis of the corpus callosum and may lead to such serious and lethal congenital malformations as holoprosencephaly or hydranencephaly [24]. This indirect evidence demonstrates the functional significance of the septum pellucidum in the central nervous system.

### Study Purpose

The purpose of this study was to analyze the risk factors for sustained positive, negative, and other psychopathological symptoms during a hospital stay.

## 2. Material and Methods

In this retrospective study, we reviewed the medical records of patients hospitalized in general psychiatric wards of the Department of Psychiatry (Medical University of Warsaw teaching hospital) and the Nowowiejski Hospital in Warsaw in the years 2006–2017. Initially, we reviewed 600 medical records of consecutively hospitalized patients. The main inclusion criterion was a discharge diagnosis of schizophrenia. 

During this initial stage, 262 patients were excluded for the following reasons: -another diagnosis, such as schizoaffective disorder or bipolar affective disorder, was established prior to the evaluated hospitalization, -unavailable neuroimaging scan data.

Once the exclusion criteria were applied, the medical records of the remaining 338 patients were analyzed during further stages of the study.

The following variables were analyzed: demographic data,information obtained via standardized history-taking,psychopathological symptoms,neuroimaging scan analysis for the presence of the CSP.

The analysis of medical records and CT scans was performed by a specialist in both psychiatry and human anatomy. The severity of symptoms was not evaluated; instead, we adopted a dichotomous system, considering the symptoms as either present (1) or absent (0).

### 2.1. Patient Demographic Data

The mean age of the 338 patients whose medical records were analyzed was 42.7 years, with a standard deviation (SD) of ±15.1 (median 40.0 years; range 18.0–83.0 years).

The mean duration of schizophrenia was 13.6 years, SD ± 12.6 (median 9.0 years; range 1–55 years) (Table 1). 

The mean duration of hospital stay was 66.1 days, SD ± 38.0 (median 57.0 days; range 5–269 days). On admission, each patient’s initial and definitive diagnosis was paranoid schizophrenia. The study population comprised 179 women (52.96%) and 159 men (47.04%). 

### 2.2. Number of Hospitalizations

A total of 249 patients (73.67%) were admitted to the hospital voluntarily, and 89 (26.33%) were admitted against their will. For 37 patients (10.95%), this was the first hospitalization at a psychiatric hospital, whereas 301 patients (89.05%) had been hospitalized before, with 135 patients (39.94%) having been hospitalized three to six times, and in the case of 110 patients (32.54%) this had been their first or second hospitalization. A total of 93 patients (27.51%) had been hospitalized more than 6 times. 

### 2.3. Comorbidity

A total of 309 patients (91.42%) were discharged with the sole diagnosis of paranoid schizophrenia, whereas 29 patients (8.58%) were discharged with an additional diagnosis of another mental condition. 

Thirty-three patients (9.76%) had a positive history of alcohol dependence; 46 patients (13.61%) had a substance abuse disorder; 15 patients (4.44%) had a history of benzodiazepine dependence. Twenty-seven patients (7.99%) had a history of head injury; 20 patients (5.92%) had had a loss of consciousness; and two patients (0.59%) had a history of seizures (of unknown etiology). Ninety-six patients (28.40%) had a history of surgical procedures, and 108 patients (31.95%) had a history of somatic disease. Our analysis revealed the following conditions in the evaluated population: hypertension, diabetes, ischemic heart disease, hyperthyroidism, and hypothyroidism. Out of 48 patients (14.20%) with a history of suicide attempts, 36 patients (10.65%) had had one, 10 patients (2.96%) had had two, and two patients (0.59%) had had three suicide attempts. 

### 2.4. Family History

Fifty-eight patients (17.16%) had schizophrenia in their family history, even though the relatives considered for this particular variable were limited to vertical first-line relatives and siblings. Twenty-three patients (6.80%) had a family history of other mental illnesses. Forty-seven patients (13.91%) had a family history of alcohol dependence, and three patients (0.89%) had a family history of drug dependence. Seventeen patients (5.03%) had a relative (ranging from first to third degree) who had committed suicide. 

### 2.5. Psychopathological Symptom Characteristics

The symptoms evaluated in the initial analysis were divided into three groups: positive, negative, and other psychopathological symptoms (DSM IV-TR). Positive symptoms included delusions, hallucinations, and disorganized behavior. Negative symptoms included emotional blunting, abulia, anhedonia, autism, and alogia. Other, associated (though atypical) symptoms of schizophrenia included mood disorders, incoherent trains of thought, anxiety, cognitive dysfunction, somatic disorders, and sleep disturbances. 

### 2.6. Neuroimaging Scan Evaluation for the Presence of the CSP

Neuroimaging scans (CT scans and the associated radiologist reports) of the patients participating in the study were evaluated retrospectively for the presence of the CSP (Figure 3) [26].

The neuroimaging scans were performed for clinical indications with the use of a Philips Brillance 64 channel CT scanner, with the following, X-ray source-dependent image acquisition parameters: 120 kV, 140 mA and a scan cycle time of 0.4 s. The remaining parameters that affect image quality were as follows: collimation 64 × 0.625 mm, acquisition field of view (FOV) 250 mm, 1024 × 1024 pixel matrix, pixel size 0.24 mm. The acquired data were reconstructed, presented in the DICOM format, and evaluated by a radiologist with the use of an OsiriX© DICOM viewer. Available literature reports indicate that various authors who assess magnetic resonance images of the central nervous system for persistent CSP consider various values of its length, e.g., 10 mm [27], or 6 mm or more [28].

In our study, the expert who evaluated the CT scans and the associated radiologist reports confirmed the presence or absence of the CSP.

Neuroimaging scan analysis helped detect the presence of the CSP in 47 patients from the evaluated group (13.91%). 

### 2.7. Statistical Analysis

The statistical analysis of quantitative variables was conducted with the use of descriptive statistics, such as means, standard deviations, medians, and ranges. The Shapiro–Wilk test was used to test whether the distribution of the analysed quantitative variables deviated from a normal distribution. Since the distribution of the evaluated quantitative variables was shown not to be normal, groups of samples were compared with an unpaired two-sample Wilcoxon test (WilcoxonRank_Sum test). Relationships between categorical variables were evaluated with the use of contingency tables and the chi-square test or Fisher’s exact test for small sample sizes.

The logistic regression generalized linear model (GLM) was employed in multivariate analysis to evaluate a relationship between the effect of treatment (the presence of symptoms on discharge) and risk factors. The optimal model was selected based on the Akaike information criterion (AIC) statistic. Goodness-of-fit testing for receiver operating characteristic (ROC) curves was assessed with the use of the area under the curve (AUC). *p*-values of less than 0.05 were considered statistically significant. Statistical analysis calculations were conducted with the use of SAS v.14.3.

## 3. Results

Our multivariate analysis helped select the sets of variables (multivariate models) that best predicted the risk of sustained symptoms during hospitalization. Out of the analysed 338 patients whose medical records were analysed, 337 patients (99.70%) had positive symptoms on admission. At discharge, positive symptoms were still present in 208 patients (61.72% out of the 337 patients who had had positive symptoms on admission). A multivariate analysis showed that the variables associated with a higher risk of sustained positive symptoms of schizophrenia at discharge included older age, greater number of hospitalizations, being the subject of physical coercion during the evaluated hospitalization, and the presence of symptoms from all three groups on admission. The risk of sustained positive symptoms was lowered by a negative history of suicide attempts, the presence of a mood disorder on admission, and a persistent CSP. The results of multivariate analysis of the risk factors for the presence of positive symptoms at discharge are presented in Table 2 and in Figure 4.

Negative symptoms were present in 299 out of 338 patients (88.46%) on admission. At discharge, these symptoms were still present in 254 patients (84.95% of the 229 patients who had had negative symptoms on admission). A multivariate analysis demonstrated that the risk factors of sustained negative symptoms of schizophrenia at discharge were older age, past suicide attempts, the presence of negative symptoms of schizophrenia or other psychopathological symptoms on admission, and the absence of psychomotor agitation on admission. The presence of persistent CSP lowered the risk of sustained negative symptoms at hospital discharge. The results of multivariate analysis of the risk factors for the presence of positive symptoms at discharge have been presented in Table 3 and Figure 5.

Other psychopathological symptoms were present in 320 patients (94.67%) on admission; at discharge, such symptoms were still present in 205 patients (64.06% out of the 320 patients who had had other psychopathological symptoms on admission). Older age and a higher numbers of past hospitalizations increased the risk of sustained other psychopathological symptoms at hospital discharge. Other risk factors for sustained other symptoms at discharge were also shown to be a positive family history of alcohol abuse/dependence, the lack of affect dysregulation on admission, stable mood on admission, no anxiety on admission, the presence of negative symptoms, and other psychopathological symptoms on admission. The presence of persistent CSP lowered the risk of sustained other psychopathological symptoms at hospital discharge. The results of a multivariate analysis for the variables affecting the risk of other psychopathological symptoms being present at discharge have been shown in Table 4 and in Figure 6.

In summary, significant risk factors for sustained positive symptoms were older age, higher number of hospitalizations, past suicide attempts, an absent CSP, stable mood on admission; positive symptoms on admission, negative symptoms on admission, other symptoms of schizophrenia on admission, and being subjected to physical coercion during hospitalization. 

Significant risk factors for sustained negative symptoms were older age, past suicide attempts, an absent CSP, psychomotor agitation on admission, negative symptoms on admission, and other symptoms of schizophrenia on admission.

Significant risk factors for sustained other symptoms of schizophrenia were older age, a higher number of hospitalizations, a family history of alcohol dependence, an absent CSP, affect dysregulation on admission, stable mood on admission, anxiety on admission, negative symptoms on admission, and other symptoms of schizophrenia on admission.

A data analysis demonstrated that individuals with persistent CSP had higher rates of substance dependence and significantly higher rates of schizophrenia in relatives.

## 4. Discussion

Schizophrenia causes an estimated 1% increase in premature mortality and years lived with a disability (collectively known as disability-adjusted life years, DALYs). This is a relatively high proportion, considering the fact that the prevalence of schizophrenia is several times lower than that of depression or cardiovascular disease [29,30].

Neither the causes nor any adequate causal treatments for schizophrenia are currently known. The great variety of symptoms and the various extents to which patients with schizophrenia are able to function are the basis for the diversity of clinical presentations. Epidemiological data indicate that approximately 60–70% of patients with schizophrenia develop delusions, whereas 30–40% of patients develop hallucinations [31,32]. 

Retrospective studies, which involve analysing records from the past while searching for the causes of current health problems, are challenging due to the limitations posed by the research material. These limitations stem from the fact that the records are typically a collection of descriptive entries written by numerous people in various degrees of detail over a period of many years.

Our analysis of positive and other symptoms of schizophrenia remaining throughout hospitalization demonstrated emotional disturbances to be a factor potentially limiting the sustained nature of symptoms. Associating schizophrenia with affective disorders (schizoaffective disorders) is the subject of much discussion and controversy. Interpreting the results of affective disorder assessments done during ongoing acute positive symptoms is often very difficult [32,33,34]. 

Somatic conditions and schizophrenia tend to co-exist in the elderly. In 46% of these individuals, their somatic condition exacerbates their mental condition, which is life-threatening in 7% of patients [31,35,36].

Epidemiological data indicate the frequent co-existence of schizophrenia and substance abuse disorders (50–60% of patients with schizophrenia suffer from alcohol abuse, and 20–40% of patients with schizophrenia abuse other substances) [33]. 

Our study showed that alcohol abuse/dependence in a close family member increases the risk of sustained symptoms of schizophrenia that are categorized as ‘other’. This seems to support the theory of a familial/genetic predisposition to schizophrenia—which posits that the members of the patient’s family also have symptoms categorized as ‘other’, but these are compensated for by alcohol abuse [37,38]. This may be a research hypothesis for further studies.

Literature data indicate that a normal development of the septum pallucidum determines the development of the forebrain [24]. Septim pellucidum development is associated with the development of the corpus callosum [39,40]. In addition, CSP is present at 100% in the fetus, and consequently decreases to a level of about 6–7% in the non-psychotic population [41]. It was shown that persistent CSP may be a cause of serious and lethal congenital malformations of the central nervous system by leading to corpus callosum agenesis [24,39,40].

Unlike the interthalamic adhesion and the surface morphology of the orbitofrontal cortex, the CSP may be easily and reliably analysed via CT.

Liu-Xian Wang et al. [28], who analysed 25 studies conducted in 2392 patients with mental disorders and 1445 mentally healthy individuals, observed that the rates of persistent CSP were significantly higher in patients with mental disorders in comparison with those in healthy individuals. Those authors also showed comparable rates of persistent CSP in patients diagnosed with schizophrenia and in those with mood disorders.

The analysis of the data from our study showed significantly higher rates of substance abuse and relatives with schizophrenia in individuals with persistent CSP. 

## 5. Conclusions

Despite many years of research, the pathogenesis of schizophrenia remains unclear. This paper presents a novel attempt at identifying factors associated with persistent symptoms of schizophrenia considering structural variations of the central nervous system. The variety of symptoms and course patterns and the complex pathogenesis of schizophrenia have encouraged many attempts at organizing the symptoms into more distinct diagnostic categories. 

Retrospective studies, which are based on past records, are inherently limited by the heterogeneous research material, which is produced by numerous people with a various degree of detail over a period of many years.

The purpose of this study was to analyse risk factors for persistent positive, negative, and other symptoms of schizophrenia in hospitalized patients. The results of this research may need to be supported by further studies.

## Figures and Tables

**Figure 1 ijerph-20-04592-f001:**
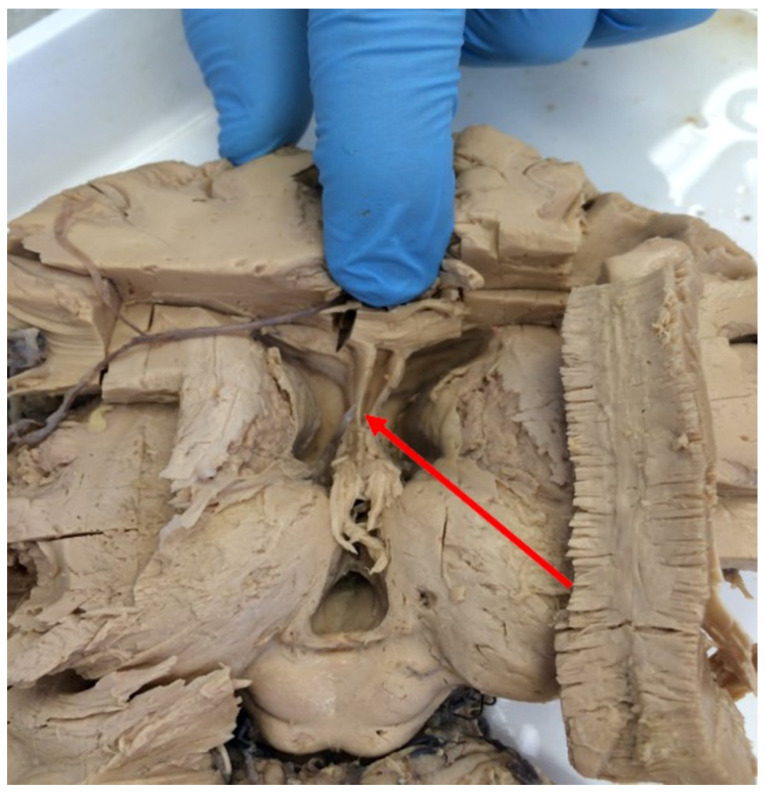
Brain specimen fixed in formaldehyde. Image courtesy of the Department of Descriptive and Clinical Anatomy, Center for Biostructure Research, Medical University of Warsaw. The arrowhead shows separated laminae of the septum pellucidum (preserved cavum septi pellucidi, CSP).

**Figure 2 ijerph-20-04592-f002:**
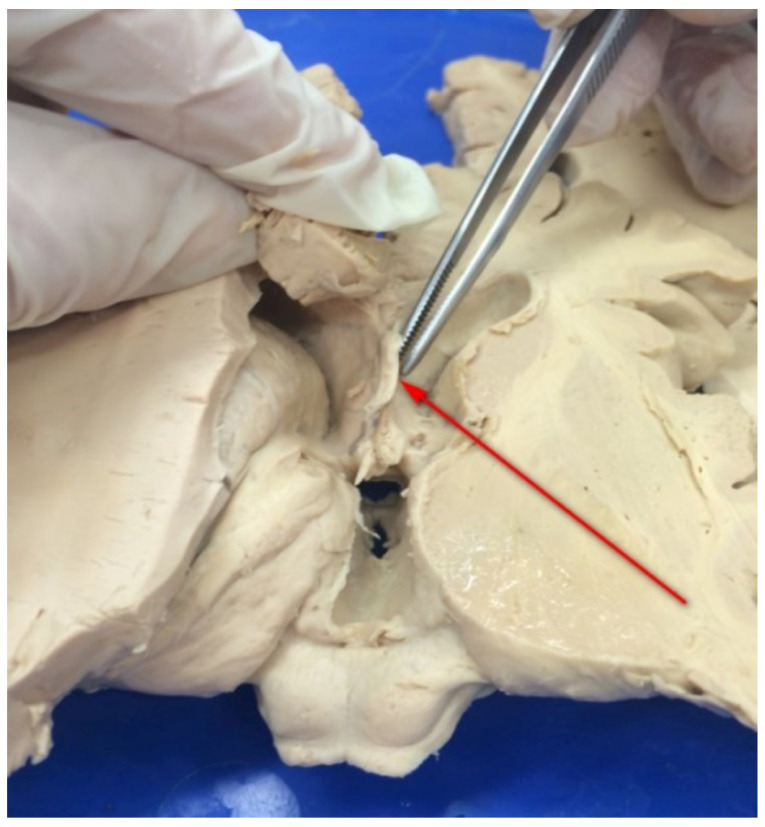
Brain specimen fixed in formaldehyde. Image courtesy of the Department of Descriptive and Clinical Anatomy, Center for Biostructure Research, Medical University of Warsaw. The arrowhead shows fused laminae of the septum pellucidum.

**Figure 3 ijerph-20-04592-f003:**
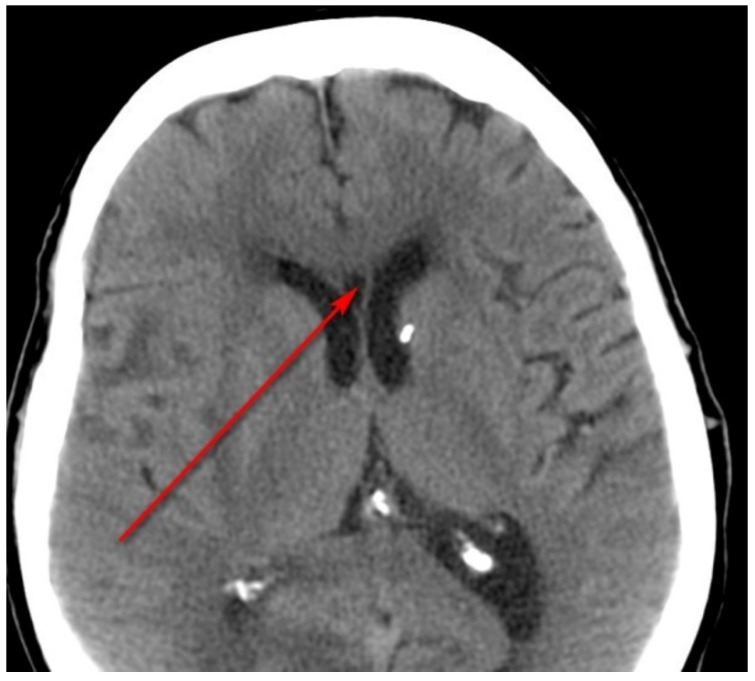
Computed tomography scan of the head of patient No. 62, a 55-year-old female. The arrowhead shows the cavum septi pellucidi (CPS).

**Figure 4 ijerph-20-04592-f004:**
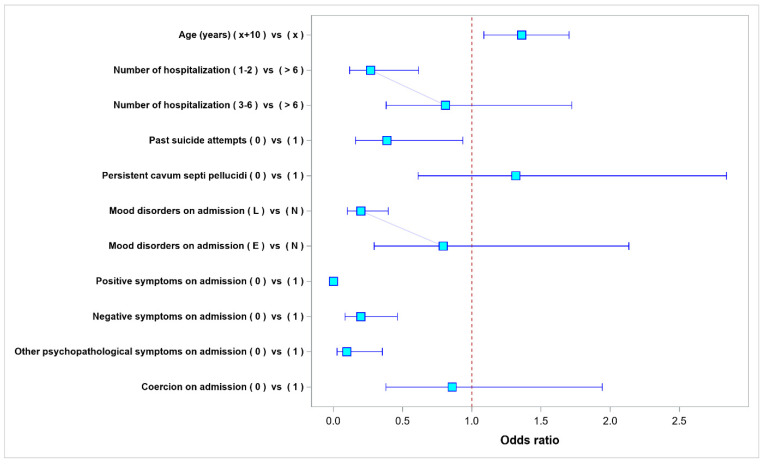
Odds ratio values for parameters affecting the presence of positive symptoms at hospital discharge.

**Figure 5 ijerph-20-04592-f005:**
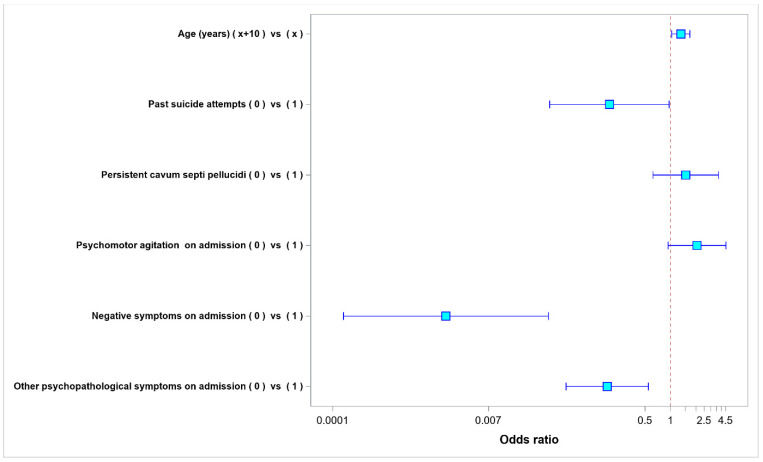
Odds ratio values for the parameters affecting the presence of negative symptoms at hospital discharge.

**Figure 6 ijerph-20-04592-f006:**
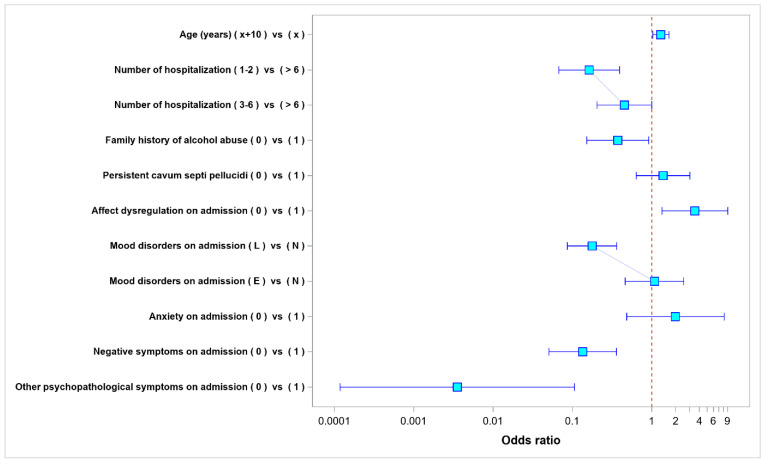
Odds ratio values for the parameters affecting the presence of other symptoms at hospital discharge.

**Table 1 ijerph-20-04592-t001:** Characteristics of the patients (N = 338) included in the study.

	Mean	SD	Median	Range	
				Minimum	Maximum
Patient age (years)	42.7	15.1	40.0	18.0	83.0
Disease duration (years)	13.6	12.6	9.0	1.0	55.0
Duration of evaluated hospitalization (days)	66.1	38.0	57.0	5.0	269.0

**Table 2 ijerph-20-04592-t002:** Multivariate model—odds ratio values for the parameters affecting the presence of positive symptoms at hospital discharge.

Odds Ratios						
Factor	Level-1	Level-2	Odds Ratio	95% Lower Confidence Limit	95% Upper Confidence Limit	*p*-Value
	(Actual)	(Base)				
Age (years)	x + 10	x	1.36	1.087	1.702	0.007
Number of hospitalizations	1–2	>6	0.27	0.118	0.616	0.002
Number of hospitalizations	3–6	>6	0.81	0.381	1.723	NS
Past suicide attempts	0	1	0.386	0.159	0.936	0.035
Persistent cavum septi pellucidi	0	1	1.319	0.612	2.84	NS
Mood disorders on admission	L	N	0.199	0.1	0.396	<0.001
Mood disorders on admission	E	N	0.794	0.295	2.134	NS
Positive symptoms on admission	0	1	<0.001	<0.001	>999	NS
Negative symptoms on admission	0	1	0.198	0.084	0.463	<0.001
Other psychopathological symptoms on admission	0	1	0.098	0.027	0.353	<0.001
Coercion on admission	0	1	0.859	0.38	1.943	NS

Note: N—normal mood, L—low mood, E—elevated mood, NS—not significant.

**Table 3 ijerph-20-04592-t003:** Multivariate model—odds ratio values for the parameters affecting the presence of negative symptoms at hospital discharge.

Odds Ratios						
Factor	Level-1	Level-2	Odds Ratio	95% Lower Confidence Limit	95% Upper Confidence Limit	*p*-Value
	(Actual)	(Base)				
Age (years)	x + 10	x	1.327	1.037	1.699	0.025
Past suicide attempts	0	1	0.19	0.037	0.965	0.045
Persistent cavum septi pellucidi	0	1	1.518	0.623	3.697	NS
Psychomotor agitation on admission	0	1	2.064	0.938	4.538	NS
Negative symptoms on admission	0	1	0.002	<0.001	0.036	<0.001
Other psychopathological symptoms on admission	0	1	0.179	0.058	0.548	0.003

**Table 4 ijerph-20-04592-t004:** Multivariate model—odds ratio values for the parameters affecting the presence of other symptoms at hospital discharge.

Odds Ratios						
Factor	Level-1	Level-2	Odds Ratio	95% Lower Confidence Limit	95% Upper Confidence Limit	*p*-Value
	(Actual)	(Base)				
Age (years)	x + 10	x	1.3	1.029	1.642	0.028
Number of hospitalizations	1–2	>6	0.163	0.067	0.394	<0.001
Number of hospitalizations	3–6	>6	0.451	0.204	0.999	0.049
Family history of alcohol abuse	0	1	0.374	0.152	0.918	0.032
Persistent cavum septi pellucidi	0	1	1.39	0.638	3.028	NS
Affect dysregulation on admission	0	1	3.5	1.347	9.098	0.01
Mood disorders on admission	L	N	0.177	0.087	0.363	<0.001
Mood disorders on admission	E	N	1.086	0.464	2.542	NS
Anxiety on admission	0	1	1.991	0.484	8.184	NS
Negative symptoms on admission	0	1	0.135	0.051	0.359	<0.001
Other psychopathological symptoms on admission	0	1	0.004	<0.001	0.106	0.001

## Data Availability

Data are available from the Department of Psychiatry, Medical University of Warsaw, 00-665 Warsaw, Poland.

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
