# Peer review of "Factors Contributing to Risk of Persistence of Positive and Negative Symptoms in Schizophrenia during Hospitalization"

_ijerph, 2023, doi:10.3390/ijerph20054592_

Round 1

Reviewer 1 Report

There are some positive and problematic elements of the current paper.  In general, the writing is clear and the results are well organized. However, the main challenge I had was fully understanding the inclusion of CSP. For example, the authors do not explicitly state it's connection to schizophrenia or mental health outcomes in the Introduction.  Further, there are no a priori predictions about what outcomes might be anticipated (related to the CSP) in their sample.  I'm presuming that there must have been something that motivated the examination of this region while the study was conceptualized because many participants were excluded if they did not have a neuroimaging scan.  Similarly, they provide three figures dedicated to the location of this region without any additional data-interpretative value (e.g., image showing a composite image of those with persistent symptoms vs. those with non-persistent symptoms). 

Some of the key research variables are not clearly operationalized. For example, the symptoms (positive, negative, other) are listed. But there does not appear to be any clinical rating of severity (despite the authors mentioning the SAPS/SANS in the Introduction). Thus, it is not clear how to interpret the analyses that involve "mood disorder on admission" or "positive symptoms on admission" as compared to symptoms on discharge.  According to the paper, 99.7% had positive symptoms on admission, and 61.72% of patients had positive symptoms at discharge. How was this recorded in the charts? (e.g., check list? narrative discharge notes? Who is providing the ratings at these time points?). 

Similarly, I had some difficulty matching the data presented in the Tables to the narrative description provided in the results (e.g., all rows related to "persistent cavum septi pellucidi" indicate non-significant findings, while reference to these findings in the written portion of the Results suggest a relationship: "The risk of sustained positive symptoms was lowered by...persistent CSP. ..The presence of persistent CSP lowered the risk of sustained negative symptoms... The presence of persistent CSP lowered the risk of sustained other psychopathological symptoms".  Additionally, when discussing the CSP findings in the Discussion portion of the paper the authors discuss previous literature (something that would be better suited for the Intro since it is the one instance where they provide a potential link of CSP to schizophrenia), but then do not discuss their own findings (or lack of findings). The provide one quick sentence indicating the relationship of substance abuse in relatives of those with persistent CSP, but provide no elaboration if this is clinically relevant.

On the positive side, this is a unique sample and potential to contribute knowledge to the field. I have provided my comments to encourage the reorganization/clarification of important elements.

Author Response

Reviewer 1

Thank you very much for your comments. These comments were very helpful while preparing the revised version of this manuscript. Here is our response to your critique:

Comments and Suggestions for Authors

  1. There are some positive and problematic elements of the current paper.  In general, the writing is clear and the results are well organized. However, the main challenge I had was fully understanding the inclusion of CSP. For example, the authors do not explicitly state it's connection to schizophrenia or mental health outcomes in the Introduction.  Further, there are no a priori predictions about what outcomes might be anticipated (related to the CSP) in their sample.  I'm presuming that there must have been something that motivated the examination of this region while the study was conceptualized because many participants were excluded if they did not have a neuroimaging scan.  Similarly, they provide three figures dedicated to the location of this region without any additional data-interpretative value (e.g., image showing a composite image of those with persistent symptoms vs. those with non-persistent symptoms). 

Response

The required information, explaining our selection of the cavum septi pellucidi (CSP) as the evaluated CNS structure, was added to the Introduction and Discussion sections, along with the relevant references.

Developmental abnormalities of midline prosencephalic structures have been linked with schizophrenia for a number of years. Although, reduction in the long diameter of the interthalamic adhesion (adhesio interthalamica) and disturbances in surface morphology of the orbitofrontal cortex seem well established in schizophrenic patients, the role of the CSP is still unclear. There are reports that negate the significance of this structure in schizophrenia, whereas some strongly support this idea.

These discrepancies focused our attention on the CSP in schizophrenic patients.

Unlike the interthalamic adhesion or surface morphology of the orbitofrontal cortex, the CSP may by easily and reliably analyzed via CT.

The following references were added to the list:

1.Brain neurodevelopmental markers related to the deficit subtype of schizophrenia.

Takahashi T, Takayanagi Y, Nishikawa Y, Nakamura M, Komori Y, Furuichi A, Kido M, Sasabayashi D, Noguchi K, Suzuki M. Psychiatry Res Neuroimaging. 2017 Aug 30;266:10-18. doi: 10.1016/j.pscychresns.2017.05.007. Epub 2017 May 20.

  1. Cavum septi pellucidi in first-episode schizophrenia and first-episode affective psychosis: an MRI study.

Kasai K, McCarley RW, Salisbury DF, Onitsuka T, Demeo S, Yurgelun-Todd D, Kikinis R, Jolesz FA, Shenton ME. Schizophr Res. 2004 Nov 1;71(1):65-76. doi: 10.1016/j.schres.2003.12.010.

  1. Some of the key research variables are not clearly operationalized. For example, the symptoms (positive, negative, other) are listed. But there does not appear to be any clinical rating of severity (despite the authors mentioning the SAPS/SANS in the Introduction). Thus, it is not clear how to interpret the analyses that involve "mood disorder on admission" or "positive symptoms on admission" as compared to symptoms on discharge.  According to the paper, 99.7% had positive symptoms on admission, and 61.72% of patients had positive symptoms at discharge. How was this recorded in the charts? (e.g., check list? narrative discharge notes? Who is providing the ratings at these time points?). 

Response

Taking into consideration your suggestion (a particularly valid one, in our opinion) we added the information on the methods of assessing the patients’ clinical condition.

Analysis of medical records and CT scans was performed by a specialist in both psychiatry and human anatomy. The severity of symptoms was not evaluated, instead, we adopted a dichotomous system, considering the symptoms as either present (1) or absent (0).

  1. Similarly, I had some difficulty matching the data presented in the Tables to the narrative description provided in the results (e.g., all rows related to "persistent cavum septi pellucidi" indicate non-significant findings, while reference to these findings in the written portion of the Results suggest a relationship: "The risk of sustained positive symptoms was lowered by persistent CSP. The presence of persistent CSP lowered the risk of sustained negative symptoms.The presence of persistent CSP lowered the risk of sustained other psychopathological symptoms".

Response

In the manuscript, the authors used the technique of variable selection based on the Akaike information criterion (AIC). While significance criteria are typically used to include or exclude variables from a model, informative criteria focus on selecting a model from a set of plausible models. Including more variables in the model slightly increases the fit of the model expressed in terms of the model probability. Unfortunately, such a fit is not desirable, so information criteria have been developed to avoid this spurious model fit leading to the selection of more complex models. Although, at first glance, selection based on information criteria seems different from selection based on the significance of variables, there is a connection between the two concepts. However, there remains a difference in the interpretation of the "significance" of the variable.

  1. Additionally, when discussing the CSP findings in the Discussion portion of the paper the authors discuss previous literature (something that would be better suited for the Intro since it is the one instance where they provide a potential link of CSP to schizophrenia), but then do not discuss their own findings (or lack of findings). The provide one quick sentence indicating the relationship of substance abuse in relatives of those with persistent CSP, but provide no elaboration if this is clinically relevant.

Response

As mentioned above, we added the information explaining our selection of the CSP as the evaluated CNS structure, along with the relevant references.

Thank you very much for thoroughly reviewing our manuscript and for providing suggestions that will certainly be helpful in extending this study further.

Reviewer 2 Report

Why use only CSP from the available CT scans, why not other brain areas?

I understand that positive and negative symptom assessment is more convenient. However, cognitive symptoms are a hallmark of schizophrenia (i.e. "Dementia praecox") and should have been investigated. Maybe some other time...

Manuscript lacks limitations/strenghts section.

From the text, I did not gather if the authors used correction for multivariate analysis. If they did, did the significance hold after correction?

Figures 1 and 2 are a nice touch. However, not that essential.

Tables 2, 3, and 4 can be joined in one table.

Author Response

Reviewer 2

Thank you very much for your comments. These comments were very helpful while preparing the revised version of this manuscript. Here is our response to your critique:

Comments and Suggestions for Authors

1.Why use only CSP from the available CT scans, why not other brain areas?

Response:

The required information, explaining our selection of the cavum septi pellucidi (CSP) as the evaluated CNS structure, was added to the Introduction and Discussion sections, along with the relevant references.

Developmental abnormalities of midline prosencephalic structures have been linked with schizophrenia for a number of years. Although, reduction in the long diameter of the interthalamic adhesion (adhesio interthalamica) and disturbances in surface morphology of the orbitofrontal cortex seem well established in schizophrenic patients, the role of the CSP is still unclear. There are reports that negate the significance of this structure in schizophrenia, whereas some strongly support this idea.

These discrepancies focused our attention on the CSP in schizophrenic patients.

Unlike the interthalamic adhesion or surface morphology of the orbitofrontal cortex, the CSP may by easily and reliably analyzed via CT.

The following (two) references were added to the list:

1.Brain neurodevelopmental markers related to the deficit subtype of schizophrenia.

Takahashi T, Takayanagi Y, Nishikawa Y, Nakamura M, Komori Y, Furuichi A, Kido M, Sasabayashi D, Noguchi K, Suzuki M. Psychiatry Res Neuroimaging. 2017 Aug 30;266:10-18. doi: 10.1016/j.pscychresns.2017.05.007. Epub 2017 May 20.

  1. Cavum septi pellucidi in first-episode schizophrenia and first-episode affective psychosis: an MRI study.

Kasai K, McCarley RW, Salisbury DF, Onitsuka T, Demeo S, Yurgelun-Todd D, Kikinis R, Jolesz FA, Shenton ME. Schizophr Res. 2004 Nov 1;71(1):65-76. doi: 10.1016/j.schres.2003.12.010.

2.I understand that positive and negative symptom assessment is more convenient. However, cognitive symptoms are a hallmark of schizophrenia (i.e. "Dementia praecox") and should have been investigated. Maybe some other time...

Response:

Thank you very much for this suggestion. We will include cognitive symptom assessment in the extension of this study.

3.From the text, I did not gather if the authors used correction for multivariate analysis. If they did, did the significance hold after correction?

Response:

The selection of the model was carried out on the basis of the Akaike information criterion (AIC). Then, the tests were carried out in accordance with the methodology of the regression model (Generalized Linear Model), taking into account the multidimensional structure. This made it possible to implicitly take into account the problems associated with multiple testing corrections

  1. Figures 1 and 2 are a nice touch. However, not that essential.

Tables 2, 3, and 4 can be joined in one table.

Response:

We believe that providing pictures of the anatomical structures discussed in the text increases overall clarity of the manuscript.

Tables 2, 3, and 4 show the results obtained with the multidimensional models presented in graphs. We believe that combining this data into one big table would make it less clear.

Round 2

Reviewer 1 Report

This is a review of the revised manuscript titled Factors Contributing to Risk of Persistence of Positive and Negative Symptoms in Schizophrenia During Hospitalization. The authors have added information in the Introduction portion of the paper that clarify the potential relevance of CSP to schizophrenia, including two new citations, and discussion of three different approaches to understanding etiology (i.e., monogenous, polygenous, and heterogenous-- citations still  needed).  These additions are positive, yet there are portions of the Intro that leave me concerned. I had not specifically mentioned in my last review since I focused on major omissions and had recommended a 'rejection' given the significant need for changes beyond the scope of a revise-resubmit. Regardless, the concerns remaining in the Intro include the loose citation of sources supporting main ideas. For example, citations 3 & 4 are used to support the notion that schizophrenia can be chronic (and progressive) for a portion of those with the disorder. This is not necessarily a controversial statement, but the the citations are from the early 2000s and not clearly connected to this idea. One is a book on the cognitive treatment of schizophrenia, and one is a article comparing amisulpride and risperidone in 'chronic schizophrenia'.  Similarly, the next sentence states that "This progressive condition is characterized chiefly by maladaptive personality traits and personality disintegration...". The single citation provided for this assertion (number 5) was a small study looking at subjectively rated characteristics requiring recall of former characteristics and the impression of changes.  The actual conclusions of this study were much more limited and nuanced: "Patients indeed feel that their illness has a negative impact on their personality, and this may play a role for individual fears, subjective well-being, and quality of life". Thus, if the authors wish to retain this statement they should be better able support their ideas with appropriate references that are clearly linked to the assertions while being careful to not overstate. In another instance, the authors cite reference #12 to suggest that ICD and DSM systems "do not classify or group" symptoms as 'positive or negative, but rather characterize symptoms "first-ranked... and second-rank... symptoms".  This is just not accurate.  The DSM absolutely has a grouping for negative symptoms.  The article (citation #12) does not address the issue of 'first-ranked' and 'second-ranked' symptoms, which harkens back to Kurt Schneider's early distinctions.  Perhaps the authors are referring to the comment in the article that the diagnosis of schizophrenia must include the presence of delusions, or hallucinations, or disorganized speech. Additional clarification is certainly warranted. 

The authors provided a clarification about how the presence or absence of symptoms were rated. This was instructive, although it did make me wonder if the portion of the Intro where they discuss the symptom rating systems usually employed (e.g. SAPS/SANS) is relevant or necessary.  They could focus on providing citations for the subsequent sentence (line 80) where they state: "Apart from the positive and negative symptoms of schizophrenia, the wide variety of psychopathological manifestations includes also other symptoms, which often accompany other mental conditions. These other, associated, but atypical symptoms of schizophrenia include mood disorders, anxiety, incoherent trains of thought, cognitive dysfunction, somatic disorders, and sleep disturbances".  This section is used to create the context for their later "Other" symptom grouping. But no citations are added, nor discussion about how some of these elements are viewed as being integral symptoms of the disorder, whereas others are common co-occurrences (not truly atypical). If the authors are attempting to justify the grouping of such a collection of symptoms, they should provide some overt discussion on this (e.g., why did they not look at disorganized symptoms, or cognitive symptoms, as additional symptom groupings? These are commonly identified in the field and are included in this larger grouping). 

I appreciate the many limitations the authors are confronted with in attempting to look at factors associated with persisting symptoms using records. The level of detail in discharge notes (in particular) is extremely variable from person-to-person, and from day-to-day. So, this (at a minimum) should be acknowledged and discussed as an important limitation.  

Author Response

Reviewer 1

Thank you very much for your second round of comments, which helped improve this manuscript further. Having taken into consideration all your suggestions, we introduced the following changes to the manuscript:

We altered the set of references by removing those items that you identified as irrelevant.

We added several reference articles on topics that are directly associated with our manuscript.

The following references were added:

DeLisi, L.E.; Sakuma, M.; Tew, W.; Kushner, M.; Hoff, A.L.; Grimson, R. Schizophrenia as a chronic active brain process: a study of progressive brain structural change subsequent to the onset of schizophrenia. Psychiatry. Res. 1997, 74, 129–140. DOI: 10.1016/s0925-4927(97)00012-7.

Ho, B.C.; Andreasen, N.C.; Nopoulos, P.; Arndt, S.; Magnotta, V.; Flaum, M. Progressive structural brain abnormalities and their relationship to clinical outcome: a longitudinal magnetic resonance imaging study early in schizophrenia. Arch Gen Psychiatry. 2003, 60, 585-594. DOI: 10.1001/archpsyc.60.6.585.

Emsley, R.; Chiliza, B.; Asmal, L. The evidence for illness progression after relapse in schizophrenia. Schizophr Res. 2013, 148, 117-121. DOI: 10.1016/j.schres.2013.05.016.

The Introduction and Discussion sections were expanded by the addition of the diagnostic criteria for schizophrenia based on the DSM-IV-TR and DSM-5 systems.

The Discussion and Conclusions were expanded to include the limitations associated with conducting retrospective studies, as suggested by the Reviewer.

Thank you very much for, once again, thoroughly reviewing our manuscript and for providing suggestions that will certainly be helpful in extending this study further.
